# Repurposing N-Doped Grape Marc for the Fabrication of Supercapacitors with Theoretical and Machine Learning Models

**DOI:** 10.3390/nano12111847

**Published:** 2022-05-27

**Authors:** Kethaki Wickramaarachchi, Manickam Minakshi, S. Assa Aravindh, Rukshima Dabare, Xiangpeng Gao, Zhong-Tao Jiang, Kok Wai Wong

**Affiliations:** 1College of Science, Health, Engineering & Education, Murdoch University, Perth, WA 6150, Australia; kethaki.wickramaarachchi@murdoch.edu.au (K.W.); rukshima_mech@yahoo.com (R.D.); x.gao@murdoch.edu.au (X.G.); z.jiang@murdoch.edu.au (Z.-T.J.); k.wong@murdoch.edu.au (K.W.W.); 2Nano and Molecular Systems Research Unit, University of Oulu, Pentti Kaiteran Katu 1, 90570 Oulu, Finland; assa.sasikaladevi@oulu.fi

**Keywords:** biomass carbon, winery waste, density functional theory, nitrogen doping, activating agent, energy storage, machine learning

## Abstract

Porous carbon derived from grape marc (GM) was synthesized via carbonization and chemical activation processes. Extrinsic nitrogen (N)-dopant in GM, activated by KOH, could render its potential use in supercapacitors effective. The effects of chemical activators such as potassium hydroxide (KOH) and zinc chloride (ZnCl_2_) were studied to compare their activating power toward the development of pore-forming mechanisms in a carbon electrode, making them beneficial for energy storage. GM carbon impregnated with KOH for activation (KAC), along with urea as the N-dopant (KAC_urea_), exhibited better morphology, hierarchical pore structure, and larger surface area (1356 m^2^ g^−1^) than the GM carbon activated by ZnCl_2_ (ZnAC). Moreover, density functional theory (DFT) investigations showed that the presence of N-dopant on a graphite surface enhances the chemisorption of O adsorbates due to the enhanced charge-transfer mechanism. KAC_urea_ was tested in three aqueous electrolytes with different ions (LiOH, NaOH, and NaClO_4_), which delivered higher specific capacitance, with the NaOH electrolyte exhibiting 139 F g^−1^ at a 2 mA current rate. The NaOH with the alkaline cation Na^+^ offered the best capacitance among the electrolytes studied. A multilayer perceptron (MLP) model was employed to describe the effects of synthesis conditions and physicochemical and electrochemical parameters to predict the capacitance and power outputs. The proposed MLP showed higher accuracy, with an R^2^ of 0.98 for capacitance prediction.

## 1. Introduction

Carbon-based electrode materials are suitable for energy storage applications because they possess attractive features such as excellent chemical and thermal stability, high electrical conductivity, and a large surface area [1]. Various carbonaceous materials—such as activated carbon (AC), carbon nanotubes (CNTs), carbon aerogels, graphene, ordered mesoporous carbon, carbon composites, etc. [2,3]—have been widely reported for their double-layer capacitive behavior and potential applications in supercapacitor devices. Among the carbonaceous materials reported in the literature for electrodes, AC is the most widely used, due to its high surface area and moderate production cost [4]. The synthesis routes of the other carbon materials, with high cost and difficulties of scaling-up, are not readily successful for practical implementation. Even though graphene-related allotropes of carbon have emerged as highly conductive materials, challenges exist in formulating single-layer graphene and reducing graphene layer aggregation, limiting their use in commercial applications. The nature of the carbon precursor and the processing method highly influence the derived carbon properties, such as pore size distribution, surface functional groups, and structure. Typically, the surface area and pore size are the key factors contributing to electrochemical performance, resulting in a higher power density [5].

Conventionally, AC is produced from fossil resources such as coal [6], petroleum coke [7], and coal tar [8], which provide larger surface areas for the carbon material. Finding green and renewable energy sources has become essential with the rising scarcity of non-renewable fossil fuels, increasing environmental pollution, and climate change. In recent years, biomass materials have been considered as an eco-friendly alternative to produce porous carbon, replacing conventional AC precursors [9]. In particular, using biomass waste to produce carbon materials for energy storage can yield value-added products and address challenging waste disposal problems [10]. Various studies in the literature have shown the exemplary designs and competitive electrochemical performances of biomass-based low-cost carbons compared with conventional AC [11,12,13]. Various biomass resources—such as sugarcane [10], rice husk [14], fishbone [15], garlic skin [16], wheat straw [17], etc.—have been employed to prepare AC electrodes. Biomass can inherit its pore structure or develop porous or layered structures, depending on the chosen parameters in the carbonization and physical/chemical activation processes [18]. The carbonization and activation processes are the main steps involved in synthesizing AC. They control structural and textural factors such as morphology, specific surface area, pore size distribution, pore volume, etc., which determine the performance of the carbon electrode in energy storage applications.

On the other hand, environmentally friendly biomass precursors with green approaches to synthesizing porous carbon materials are also seen to be emerging, and can provide pragmatic improvements in the sustainable processing of energy-storing materials. Goldfarb et al. [9] proposed an integrated process for extracting biofuel from the pyrolysis of pistachio nutshells and impregnating it with KOH activation processes to produce activated carbon. This integrated process increased the biofuel yield by up to 25%, and the AC was used for electrochemical capacitor studies. McNair et al. [18] reported green binders (γ-valerolactone) and solvents (i.e., cellulose acetate, carboxymethyl cellulose) as alternatives for replacing the conventional solvents (N-methyl-2-pyrrolidone; NMP) and binders (fluorinated, such as PVDF) for membrane capacitive deionization electrodes with enhanced capacitive performance. This work aimed to reduce the environmental impacts and chemical consumption used in electrode processing. All of these studies concluded that sustainable biomass electrodes could be biodegradable.

Australia is a very competitive wine-producing country that produces ~1.2 billion liters of wine per year for domestic and global sales, using ~1.73 million metric tons of wine grape crush, yielding ~0.5 million tons of GM, which requires safe disposal [19]. The waste disposal of winery residues (otherwise termed “Grape Marc”) includes onsite dumping and settling in ponds while producing tartaric acid, low-quality wine and ethanol, and animal feed, as well as composting off-site [20]. However, proper approaches to recovering the GM to produce value-added products are yet to be developed. For instance, waste dumping provides an economical means of waste disposal, but causes environmental impacts, including spreading pests and diseases [21]. After extracting tartrate or ethanol, the waste streams remain, and must be treated to prevent groundwater pollution and foul odors. In such cases, thermal conversion of the GM into valuable products such as biochar, energy, or chemical feedstock is an alternative in terms of both environmental and economic benefits [22]. GM consists largely of lignin, which possesses a 3D network structure and higher density than other lignocellulosic biomass [1]. Thus, it can be a promising feedstock for energy storage applications. With this advantage, we aimed to exploit GM—a free and sustainable material collected from Australian wineries—to produce AC. To the best of our knowledge, repurposing of GM as a precursor with extrinsic nitrogen (N)-dopant has not been reported for the fabrication of supercapacitors.

Considering the wide range of biomass precursors available and the number of synthesis conditions, it is much more beneficial to develop a proper design protocol to choose the appropriate precursors and the experimental conditions to obtain optimized electrode performance. This drives the demand to focus on state-of-the-art technologies that could assist in the design, development, and discovery of novel materials. Compared to the traditional models and algorithms, machine learning (ML) technologies have great potential to address the challenges in optimizing energy storage materials [23]. The synthesis conditions—such as carbonizing temperature, hold time duration, activation temperature, and doping amounts—strongly influence the decomposition kinetics, determining the pore-formation mechanisms of the produced carbon. The physicochemical parameters, such as the morphology and structure of a porous material, differ with the type of raw precursor and the synthesis conditions. In addition, the electrolyte type and testing conditions also affect the electrochemical performance. Therefore, it is customary to note that many parameters are involved in tailoring the energy and power densities of the energy storage materials, and it is difficult to rationalize these parameters through experimental work [24,25,26]. From our previous work, it was found that the multilayer perceptron (MLP) model can deliver highly accurate predictions of capacitance [27]. Hence, the MLP model was employed in this work, and the model predictions were compared by varying the inputs to predict the specific capacitance and power. In summary, in this study, we investigated the production of AC from Australian GM with N-dopant for capacitor applications, using parameter-extensive experimental work and a DFT model to elaborate on the effects of the N-dopant. As a time- and cost-effective method, the ML technique was applied to predict capacitance and power delivered by the AC materials.

## 2. Materials and Methods

### 2.1. Materials

GM collected from a local winery consisted of grape marc and stalks. The as-received GM was washed with deionized water, dried at 60 °C in an oven for ~24 h, ground and sieved to a size fraction of <350 µm, and then used as the precursor for AC. Urea and dried GM were mixed before carbonization and activation to synthesize the N-doped carbon. ACS-grade potassium hydroxide (KOH) and zinc chloride (ZnCl_2_) supplied by Sigma-Aldrich were used as activating agents in the chemical activation process.

### 2.2. Synthesis of GM-Derived N-Doped AC

The diagram in Figure 1 gives a complete view of GM-derived AC using varying synthesis conditions, showing that the synthesis routes are slightly different for KOH and ZnCl_2_ activation. Powdered GM was placed in a tubular furnace and then heat treated at 600 °C for 3 h in a N_2_ atmosphere for carbonization. Carbonized GM was mixed with the activating agent (KOH or ZnCl_2_) in a solution with a mass ratio of 1:3. The mixture was kept in the oven at 80 °C overnight for dehydration. Then, the dried mixture was transferred into a crucible and activated at 800 °C for 1 h in a N_2_ atmosphere. The carbon activated by KOH was termed KAC, while that activated by ZnCl_2_ was called ZnAC_1_. This process went through two steps: carbonization, followed by activation processes. The above conditions were identical to synthesizing the nitrogen-doped carbon, except that fertilizer urea was repurposed as a nitrogen additive. Urea and dried GM were mixed at a 1:1 ratio before carbonization. The AC product was termed N-doped activated carbon (KAC_urea_). The AC was synthesized using the one-step activation method at 450 °C, with ZnCl_2_ activation termed ZnAC_2_.

After the activation process of each sample, it was allowed to cool down naturally to ambient temperature. Then, the samples were washed with HCl and DI water until the pH of the filtrate became neutral. Thoroughly washed AC was dried in the oven at 105 °C for 12 h and used for further analysis.

### 2.3. Characterization of Materials

The surface morphology of the GM and its derived AC materials was investigated via field-emission scanning electron microscopy–energy-dispersive spectroscopy (FESEM–EDS, TESCAN CLARA, AXT Pty Ltd, Warriewood, NSW, Australia). X-ray diffraction (XRD) analysis was conducted (X-ray powder diffractometer, GBC Emma Theta, AXT, Warriewood, NSW, Australia) using Cu Kα radiation (λ = 1.5418 Å) operated at 28 kV and 10 mA. The surface functional groups of the prepared ACs were detected by Fourier-transform infrared (FTIR) spectroscopy. The spectra were recorded from 4000 to 400 cm^−1^. The Raman spectra of the AC materials were recorded via a high-resolution Raman spectrometer (WITec Alpha 300RA+, WITE Pte Ltd, Singapore), and microscopy was carried out using a high-resolution system with a 532 nm Nd:YAG laser. The surface area and pore structure were characterized by nitrogen (N_2_) adsorption–desorption isotherms at 77 K, using the surface area and pore analysis (SAPA, ATA Scientific Pty Ltd, Caringbah, NSW, Australia) instrument (micromeritics, TriStar II *Plus*).

Synthesized AC materials, acetylene black, and PVDF binder were mixed at a mass ratio of 75: 15: 10 in a slurry using N-methyl-2-pyrrolidone (NMP) solvent for electrode coating. The graphite current collector (1 cm^2^ area) was coated with ~3 mg mass loading using the slurry coating method, and then dried at 80 °C for 12 h. The active mass coated on the graphite was determined by the mass weighing before and after coating. Cyclic voltammetry (CV), galvanostatic charge–discharge (GCD), and electrochemical impedance spectroscopy (EIS) tests were carried out in a three-electrode system (half-cell measurements) with Hg/HgO as one reference (for the alkaline electrolytes), Ag/AgCl as another reference (for the salt electrolytes), and a Pt wire counter electrode using a Bio-logic SP-150 potentiostat. CV measurements were carried out at different scan rates of 5–60 mV s^−1^, GCD tests were conducted at different current densities of 2–6 mA cm^−2^, and EIS was performed at a frequency range of 10 mHz to 100 kHz, with 10 mV amplitude. The equations used to calculate the specific capacitance (C_sp_, F g^−1^), energy density (E, Wh kg^−1^), and power density (P, W kg^−1^) using GCD data are given in the Appendix A.

### 2.4. Density Functional Theory (DFT) Calculations

DFT simulations were performed using the Vienna Ab initio Simulation Package (VASP) [15]. A graphite bilayer was simulated using the supercell approach. A kinetic energy cutoff of 450 eV was used to explain the plane waves included in the basis set. The exchange and correlation interactions were expanded using the projector-augmented wave (PAW) [28] within the Perdew–Burke–Ernzerhof functional (PBE). To describe the semi-empirical corrections, DFT-D3 formalism was used [29] within the general gradient approximation (GGA). A 3 × 3 × 1 bilayer graphite was simulated, and a vacuum region of 15 Å was employed along the Z direction to avoid interaction between the repeating images. A Monkhorst–Pack k-grid of 9 × 9 × 1 was used for the Brillouin zone integration [30]. The energy and force convergence criteria were set to 10^−6^ eV and 10^−3^ eV/Å, respectively.

### 2.5. Multilayer Perceptron (MLP) Model

The MLP model better extracts the correlations between inputs and outputs to predict the electrochemical performance of the energy storage carbon [27,31]. Therefore, we employed the MLP model with three hidden layers to predict the specific capacitance and power density from the collected data. Appendix A presents the ML model architecture, which consists of three hidden layers with 40, 60, and 15 nodes. The majority of the datasets were created from the present work, with the remainder sourced from our recent study [27]. In this work, we used 100 datasets based on in-house experiments conducted to synthesize waste-derived AC biomass. The electrochemical data were collected from the three-electrode experiments so as to better understand each electrode’s performance with changing current rates. The model effectively distinguishes between the different samples when many sample features are included as inputs. Firstly, to quantitatively determine the effects of the inputs, the MLP model employed used 6 input parameters, as in the previous work [27]. Then, the number of input parameters was expanded to 21 to increase the parameter contribution for capacitance prediction, and the accuracy of the models was compared. The coefficient of determination (R^2^ in Equation (S4)), mean squared error (MSE in Equation (S5)), and mean absolute error (MAE in Equation (S6)) were used as the evaluation metrics to determine the model’s accuracy [31]. In general, the R^2^ value should be close to 1 for a perfect fit for the regression models, while the MSE and MAE values should be close to 0 to indicate fewer errors in fitting the data. The smaller the MSE and the MAE, the closer the predicted and the actual experimental data. The key inputs employed in the MLP model are given in Appendix A.

## 3. Results and Discussion

### 3.1. Physicochemical Characterization of the GM-Derived N-Doped AC Materials

#### 3.1.1. Morphology Analyses

Figure 2 shows the surface morphologies of the GM and its derived AC materials. Globular particles with a fibrous structure were observed on the surface of the dried GM (Figure 2A–C). No visible pores were observed on the surface. After carbonization and subsequent activation with KOH, the KAC demonstrated a porous structure due to the removal of hemicellulose and cellulose. Twisted-shaped particles with irregular sizes varying from the sub-micrometer scale to tens of micrometers were also observed (Figure 2D–F). Interestingly, for KAC_urea_, Figure 2G–I show a flattened-shaped morphology with well-distributed porosity and an interpore connection. The decomposition of N-functionalities from the urea-mixed GM evolved into gaseous species around 520 °C [32], increasing the defect sites and mesopores of KAC_urea_ compared to KAC, as visible in Figure 2D–F. In the case of the ZnCl_2_ activating agent, ZnAC_1_ (Figure 2J–L) showed a stacked granular morphology with a very low number of pores on the surface. In comparison, ZnAC2 (Figure 2M-O) shows large cavities on the surface with some small pores embedded.

The SEM image (Appendix A) and EDS elemental mapping (Appendix A) for KAC_urea_ illustrate the element distribution of the AC surface. Appendix A illustrates a substantial amount of carbon present in the sample. The urea in the GM precursor embedded the nitrogen within the carbon structure, as seen in Appendix A. The microstructure of KAC_urea_ was examined using TEM, as shown in Figure 3A,B. The HRTEM image (Figure 3B) reveals the amorphous structure of KAC_urea_, which is consistent with reported work on hard carbon synthesized via pyrolysis of carbonaceous precursors such as pomelo peels, banana stems, and corn cobs [33,34]. This further demonstrates that the elements present in the materials are similar to the previously observed SEM and EDS images in Appendix A.

#### 3.1.2. Structural and Spectral Analyses

The XRD patterns of the GM-derived AC materials using different activating agents are given in Figure 4A. All of the samples exhibited two characteristic broad diffraction peaks of turbostratic carbon (t-carbon) positioned around 25° and 44°. The diffraction peak at around 25° was attributed to the disordered (amorphous) carbon corresponding to the 002 crystal plane. The peak at 44° was indexed to the 100 plane of graphitized carbon [35]. In contrast to ZnAC_1_ and ZnAC_2_, the KAC and KAC_urea_ samples showed a slight shift in the 002 peak to a lower angle, positioned around 22° in the XRD pattern. This implies that the “K” penetration in carbon occurs during the KOH activation, and induces higher interlayer spacing of (002) d_002_. The higher D-spacing increases the number of active sites for the electrolyte ion (adsorption–desorption) interaction during charging/discharging. The broader peak seen in the KAC_urea_ indicates that the highly amorphous nature of the sample comes from the N-functionalities on its surface. For ZnAC_1_, the 002 peak was well-defined and intense, implying a crystalline material supported by the morphology seen in the FESEM images discussed earlier. By contrast, the ZnAC_2_ sample, which was carbonized and activated at a lower temperature of 450 °C, showed less crystallinity. These observations suggest that the KOH activation gives a more disordered crystal structure, and is relatively suitable for energy storage performance.

A qualitative analysis of functional groups in the GM-derived AC materials was carried out via FTIR spectroscopy in the mid- and near-IR regions (Figure 4B). The FTIR curves of all AC spectra were similar. A broader band observed around 3400 cm^−1^ could be attributed to the O–H stretching vibration of hydroxyl functional groups [36]. This was very likely caused by the dehydration of GM during the carbonization process. The position of the bands shifted to a lower wavenumber as the carbonization temperature increased for ZnAC_1_ compared to ZnAC_2_. This could be due to the disruption of H-bonds established by OH groups. The peak at 1600 cm^−1^ was related to the C=C stretching of aromatic rings and C=O conjugated with the aromatic rings [37]. The peak around 960 cm^−1^ could be associated with stretching vibration of C-C or C-H groups [38]. Overall, the oxygen-containing moieties—including C=O functionalities on the surface of GM-derived AC materials—may significantly contribute to capacitive performance [39].

The structure, defects, and disordered nature of the carbon materials could be determined by Raman spectroscopy. The Raman spectra for GM-derived AC materials using different activating agents and N-dopants are shown in Figure 4C. The D-band around 1345 cm^−1^ was related to the sp [3] carbon sites, indicating that the defect peak arises due to the A1g mode of vibration. The nearby peak around 1580 cm^−1^ represents the G-band related to the sp [2] carbon sites, and indicates the graphitic peak [40]. The graphitic peak was observed because of the E2g mode of vibration and relevant stretching of the C-C carbon bond. The D- and G-band intensity (I_D_/I_G_) ratio was inversely proportional to the in-plane crystallite sizes of the AC [40]. The I_D_/I_G_ values of 0.99, 0.99, and 1.01 for KAC_urea_, KAC, and ZnAC_1_, respectively—but not the 0.97 for ZnAC_2_—indicated good graphitic crystallinity. For a non-defective carbon material, I_D_/I_G_ should be equal to zero. The calculated I_D_/I_G_ values for the AC samples imply that all four samples had defects in the graphitic carbon, with an amorphous character. Validations such as the shift in 2*θ* of XRD towards a lower angle and the higher I_D_/I_G_ ratio suggest increased interplanar spacing and a high disorder level occurred due to KOH activation.

#### 3.1.3. Surface Area and Pore Structure

The specific surface area and pore volume of the GM-derived AC, as obtained from N_2_ adsorption–desorption measurements, are given in Table 1. With the activation of KOH, the KAC sample exhibited a specific area value of 1128 m^2^ g^−1^. For N-doped carbon, the quantity of N_2_ adsorption by KAC_urea_ demonstrated a maximum surface area of 1356 m^2^ g^−1^. The key role of the activation process using the KOH activator is to generate pores to increase the surface area available for adsorption/desorption processes at the electrode–electrolyte interface. This is evidenced by the isotherms of KAC and KAC_urea_ shown in Figure 5, which do not manifest well-positioned hysteresis loops. KAC recorded a sharp increase in volume at low relative pressures, followed by reaching a plateau at high relative pressures, giving the behavior of typical type I isotherms of microporous materials according to the IUPAC classification [41]. A slight difference is seen in the KAC_urea_ isotherm; this could be due to an increased mesopore proportion under the influence of urea as a dopant. It could also be related to a combination of type I and IV isotherms, as reported in the literature [32]. Furthermore, from KAC to KAC_urea_, the pore structure changes from micropores to hierarchical pores, which are more favorable for better electrochemical performance.

The surface area values obtained for the ZnAC samples were lower than those of the KAC and KAC_urea_ samples. ZnAC_1,_ activated at 800 °C using a pre-carbonized sample, gave the lowest surface area of 29.14 m^2^ g^−1^. There was no hysteresis loop between the adsorption–desorption isotherms of ZnAC_1_ (Figure 5), possibly due to the large micropore fraction [42]. This was further verified by the micropore area percentage, which was >89%, as calculated from the data in Table 1. A larger proportion of micropore-specific surface area lowers the mesopore and macropore surface areas. Due to this outcome, the GM was directly activated at a lower temperature of 450 °C [43]. Although this increased the surface area compared to that of ZnAC_1_, the micropore area and volume remained noticeably lower, making ZnAC_2_ less suitable for capacitor fabrication. ZnAC_2_ showed an obvious hysteresis loop characteristic of a type IV isotherm for mesoporous materials [41]. It can be concluded that choosing an effective synthesis route and a suitable activating agent determines the effective pore structure through the chemical activation process.

In summary, as derived from the physical characterization data, the KAC, KAC_urea,_ and ZnAC_1_ synthesized under similar thermal conditions had distinct characteristics due to different activation mechanisms. KOH reacts with the GM-derived carbon via KOH etching, metallic K penetration, and CO/CO_2_ release to create a porous structure during heat treatment. Reactions R1, R2, R3, and R4 in the Appendix A explain the reaction mechanisms in the presence of the KOH activating agent. The synthesis conditions used here with the KOH activating agent are favorable for high-performance supercapacitor applications. The ZnCl_2_ evaporates above its boiling point of 732 °C, thereby causing imperfection of the structural and textural properties at the activation temperature of 800 °C [44]. Considering the results of both ZnAC_1_ and ZnAC_2_, the ZnCl_2_ might conduce better activation of GM between 450 °C and 800 °C. Therefore, a suitable activating agent and synthesis route are essential for pore evolution by micropore formation and pore-widening mechanisms [45]. The FESEM images also confirm that the ZnAC_1_ is highly dense, with no visible open pores on the surface, which would impede the electrolyte ion movements into the electrode structure, plausibly decreasing the electrochemical performance. Additionally, the N-dopant present in GM could enhance the surface chemistry of the KAC_urea_. The KOH activation is usually accompanied by nitrogen release, leading to a reduced amount of nitrogen. Therefore, it is difficult to obtain higher porosity along with a considerable amount of nitrogen functional groups in KAC [46]. After the carbonization step, the nitrogen functional groups could predominantly present as pyridine-N, pyrrolic-N, imine, amine, amide, quaternary-N, and pyridine-N-oxide groups on the surface of the carbon [47]. During activation, the reactions occur to expand the carbon network, forming many more micropores and mesopores in the carbon material while reducing carbon and most nitrogen. In addition, compared to KAC, KAC_urea_ favors the metallic “K” for easy access to the bulk of GM-derived carbon through newly formed channels that enhance the porous nature of carbon [47]. The role of N-functionality on the carbon surface is postulated to be pyridinic-N, and pyrrolic-N can contribute to the pseudocapacitance in alkaline electrolytes. At the same time, quaternary-N and pyridine-N-oxide can improve the charge transfer across the electrode–electrolyte interface and enhance the capacitance. In Section 3.3, on the DFT modeling results, we discuss the N-dopant effect in its most stable form of the pyridine-N-oxide present in the carbon structure.

### 3.2. Supercapacitor Applications of GM-Derived N-Doped AC: Electrochemical Performance (CV, GCD) of the Single Electrode and Symmetric Capacitor Device

The CV curves of the KAC and KAC_urea_ samples (Figure 6A,B) encircle much larger areas and give higher current responses. The curves are nearly rectangular due to predominant EDLC charge storage, while the deviations are caused by pseudocapacitance from N-functionalities. Using the GCD measurements (Figure 6C,D) at a 2 mA current rate, the calculated specific capacitances of KAC and KAC_urea_ are 117 F g^−1^ and 139 F g^−1^, respectively, which are comparable with the reports in the literature [48,49,50]. The approximately symmetrical triangular shape of the GCD curves can be observed due to EDLC and pseudocapacitor charge-storage mechanisms, which are consistent with the CV measurements.

The shape of the CV curves is retained with an increase in the scan rate from 5 to 50 mV s^−1^, showing the stability of KAC_urea_. From the calculated capacitances, KAC_urea_ performs best, which is ascribed to both the KOH activator along with the N-dopant and the development of a hierarchical pore structure during thermal treatment. With an increase in the current rate from 2 to 10 mA, the specific capacitance of KAC_urea_ decreased gradually from 139 F g^−1^ to 125 F g^−1^ due to limited access available for the electrolyte to flow into the electrode, along with the conductance resistance [51]. The CV and GCD profiles of ZnAC are illustrated in Appendix A. The CV curves displayed by the ZnAC_1_ sample (Appendix A) possess a narrow area under the curve, in elliptical shapes, and give lower current responses. The calculated specific capacitance of ZnAC_1_ was 29 F g^−1^ at a 2 mA current rate. On the other hand, ZnAC_2_ exhibited pseudocapacitance characteristics, displaying two redox pairs A/A’ and B/B’, as shown in Appendix A. The functionalization of the carbon surface, heat-treated at an elevated temperature, could enhance the faradaic contributions to the observed capacitance. However, the lower pore volume with no improvement in pore size distribution reduced the interconnection between carbon particles, resulting in a lower specific capacitance of 41 F g^−1^ at 2 mA. The GCD curves of ZnAC_1_ and ZnAC_2_ are given in Appendix A. Similar to the observed CV behavior, the curves are not triangular, and do not show many reversible characteristics. A comparison of CV, GCD, and rate performance is given in Figure 7A,B. The CV is measured at a 20 mV s^−1^ scan rate, while GCD is measured at a 3 mA current rate. The elliptical-shaped CV of the ZnAC_1_ sample gives the lowest charge storage, whereas KAC_urea_ shows the optimal charge storage, which matches the GCD measurements in Figure 7B well. Finally, we compared the energy storage capability of all of the samples based on their specific capacitances at different current rates. The retained percentages of the SCs for ZnAC_1_, ZnAC_2_, KAC, and KAC_urea_ were 30%, 66%, 80%, and 90%, respectively, as with the rate performances plotted in Figure 7C. The capacitance retention was also increased for the KOH activated carbon sample. This corresponds to the hierarchical pore formation accompanied by the defects and porous surface morphology induced by doping, along with increased electrode/electrolyte wettability of KAC_urea_. The specific capacitance of the best-performing KAC_urea_ is compared with the reported values for the biomass-derived carbon in the literature in Figure 7D. Further comparison between different parameters was conducted for the carbon obtained from various precursors, as tabulated in Table 2.

The commercial AC was characterized under similar operating conditions (Appendix A). The specific capacitance given by KAC_urea_ (139 F g^−1^) was superior to the performance of commercial AC (80 F g^−1^) tested under identical conditions, indicating its potential value for practical applications. KAC_urea_, which possesses the largest SC, was studied in different aqueous electrolytes: two alkaline electrolytes with different cations, i.e., NaOH and LiOH, and a salt electrolyte with a different anion, NaClO_4_, all with a 2 M concentration (Appendix A). CV measurements were taken at 20 mV s^−1^ (Appendix A) and the GCD was determined at a 3 mA current rate (Appendix A) in a 0V to −1V potential window. Appendix A shows the highest capacitance retention given by NaOH. A detailed discussion of the electrolyte performance is given in the Appendix A.

The symmetric capacitor device was tested by coupling two electrodes of the best-performing KAC_urea_ material. The CV and GCD profiles of the symmetric capacitor recorded at a 1 V potential window are given in Figure 8A,B, respectively. The shapes of the curves are nearly symmetrical, and the reversible behavior of the cell can be observed in all tested conditions of varied scan rates and current rates. The discharge cell capacitances calculated at current rates of 2, 3, 4, 5, and 6 mA were 47, 44, 41, 38, and 37 F g^−1^, respectively. Furthermore, the capacitance retention of ~84% and the cell efficiency of ~96% for 10,000 cycles at 5 mA are displayed in Figure 8C.

### 3.3. Theoretical Insights: Density Functional Theory (DFT) Study

To validate and bring further insights into the experimental findings, first-principles density functional theory (DFT) simulations were performed. Nitrogen-doping in GM can effectively tune the electrical properties of carbon. The optimized lattice parameters of the graphite bilayer supercell were a = b = 2.464 Å, and c = 21.711 Å. Furthermore, the following geometries were simulated: a N atom was substituted for the C atom on the surface, O adsorbate on the pristine graphite, and O adsorbed on the N-atom-substituted graphite. The optimized geometries are shown in Figure 9.

The bond distance between the O atom and graphite surfaces was calculated, and is presented in Table 3. It can be noted that the bond distance for pristine graphite and the O atom is 2.27 Å, and this bond length decreases to 1.40 for N-doped graphite. This reduced bond distance indicates that the chemisorption of the O atom onto the graphite surface is enhanced by the presence of the N-functionality on the atomic surface. The C–N bond length also elongates in the presence of O adsorption. Bader charge analysis was carried out to understand the charge-transfer properties on the surfaces, and the results are presented in Table 3. On the graphite surface, the O adsorption occurs as a result of charge gain by the O atom, and is enhanced by the presence of the N atom (−0.35 to −0.49), indicating stronger chemisorption. The N atom gains electrons from surrounding C atoms, and the charge gain decreases when the O atom is adsorbed on the surface. On the other hand, the C atoms directly bonded or in the nearest-neighbor position of the N atom lose charge to compensate for charge redistribution. Electronic structure analysis was carried out to understand the hybridization of orbitals at the surface. The density of states (DOS) of the pristine and the N-doped graphite surfaces is presented in Appendix A. In the presence of the N atom, the total DOS at the Fermi level increased, indicating that chances of hybridization with an adsorbate increase. The projected density of states (PDOS) calculated for O adsorbed on pristine and N-doped graphite surfaces is shown in Appendix A, respectively. The DOS shows metallic behavior, and is dominated by the *p* states of C and O. The presence of the N-dopant introduces additional states near the Fermi level, and the probability of hybridization of O *p* states and the graphite surface evidently increases. We calculated the difference in charge density caused by the adsorption of O at the N-doped graphitic surface. Figure 10 shows the charge density difference plot. The yellow and blue isosurfaces indicate charge gain and depletion, respectively. It can be seen that charge gain at the site of N-doping was larger, while the doped O atom also gained charge partially. The C atoms on the surface, which are directly bonded with the N atom, lose charge, and a charge reconstruction occurs at the surface, which helps with the enhanced adsorption of the O atom. The charge density difference analysis and calculated Bader charges are consistent with the shortened bond distances in the presence of N-doping, as presented in Table 3. The DFT investigations thus support the concept of N-doping, which very likely occurred during the heat treatment of the GM precursors for carbon.

### 3.4. Machine Learning Insights: MLP Model

Machine learning can dramatically accelerate calculations, improve energy storage prediction accuracy, and make optimized decisions based on comprehensive status information to develop novel materials. Machine learning prediction has also been widely adopted in the field of materials science [63,64]. Depending on its end applications, ML has been classified into material property prediction, the discovery of novel materials, and various other purposes, such as battery management, etc. When constructing the ML models for material property predictions, researchers have extended their insights on various aspects, such as predicting the electronic properties of inorganic crystals [65], geometric features of gas storage and separation using metal–organic frameworks [66], transport properties in granular materials [67], and the effect of the pore size on capacitance [68]. Moreover, ML models are also used in efficient optimization by designing their parameters [69], applying them to DFT problems [70], monitoring the state of the battery during operation [71], etc. However, surface morphology has not been considered in previous ML studies. The particle morphologies can be quite diverse, based on biomass precursor or synthesis conditions that affect the energy storage performance [72,73,74]. As a step towards extracting useful information from electrode material texture, two input features—porosity and surface pore size—were extracted from microscope images using MATLAB [75,76]. A statistical overview was obtained through Pearson correlation, as presented in Appendix A. The positive and negative correlations are represented by dark blue and red colors, and the colors in the middle region show no or weak correlations. A comparison of the MLP model using similar inputs to the previous work and the improved number of inputs in the present work is given in Figure 11A,B. It is shown that the correlation between the actual and predicted capacitances was largely improved from 0.78 (Figure 11A) to 0.98 (Figure 11B) for R^2^ when including more parameters to represent the different samples. In addition, the RMSE value decreased from 14.45 to 4.29, indicating the importance of the additional parameters included in modelling the system. The improved model was used to predict the power density, as shown in Figure 11C, which gives 0.97 for R^2^ and a value of 0.28 for RMSE. The diagonal line in the figures represents the perfect correlation between experimental values and the predictions. The results indicate that having more material informatics in the model is essential to the exploration and design of high-performing materials. This model could accurately retrieve the nonlinear patterns to correlate the inputs and outputs compared to the traditional theoretical models, and could speed up these processes in the materials science field.

## 4. Conclusions

The sustainable GM was effectively tailored into AC, making it a promising material for supercapacitor applications. The physicochemical and electrochemical results showed that KOH activation is preferable to ZnCl_2_ activation under the tested sequence of the synthesis process. The GM-derived AC (KAC_urea_) from the precursor of mixed GM and urea for N-doping further improved the physicochemical properties and, hence, the electric double-layer capacitor (EDLC) behavior. KAC_urea_ presented hierarchical pores and the highest specific surface area among all of the samples studied. Consequently, KAC_urea_ delivered the highest specific capacitance of 139 F g^−1^ at the 2 mA current rate in the alkaline NaOH electrolyte. Therefore, the symmetric capacitor behavior was studied using the best-performing KAC_urea_ electrodes in 2 M NaOH with a 1 V potential window. DFT calculations indicated that N-doping enhanced the adsorption of O atoms on the graphitic surface, owing to the charge-transfer mechanism. The electronic structure analysis revealed increased hybridization of orbitals of the adsorbate and graphite surface in the presence of the N-dopant. Therefore, improved ion adsorption occurs during the charge–discharge process. Increasing the number of inputs in the MLP model delivered the best correlations between actual and predicted capacitances. This study could support future efforts in interdisciplinary areas such as electrochemistry, materials science, and energy, combined with ML applications for energy storage devices.

## Figures and Tables

**Figure 1 nanomaterials-12-01847-f001:**
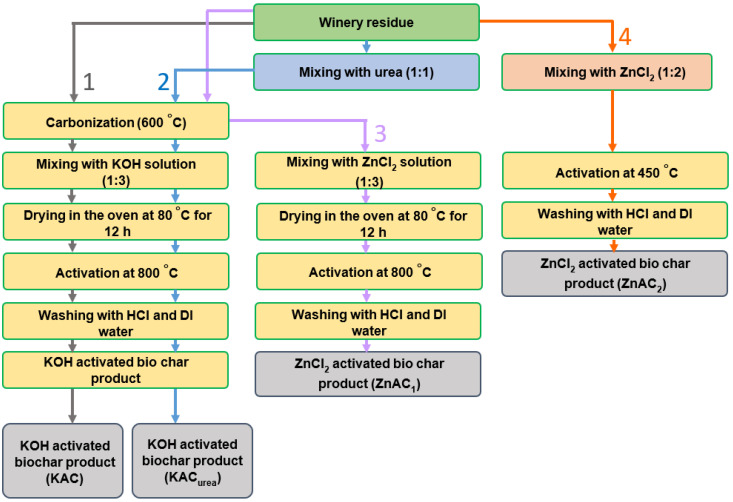
Flow diagram of the synthesis conditions followed in producing activated carbon (AC).

**Figure 2 nanomaterials-12-01847-f002:**
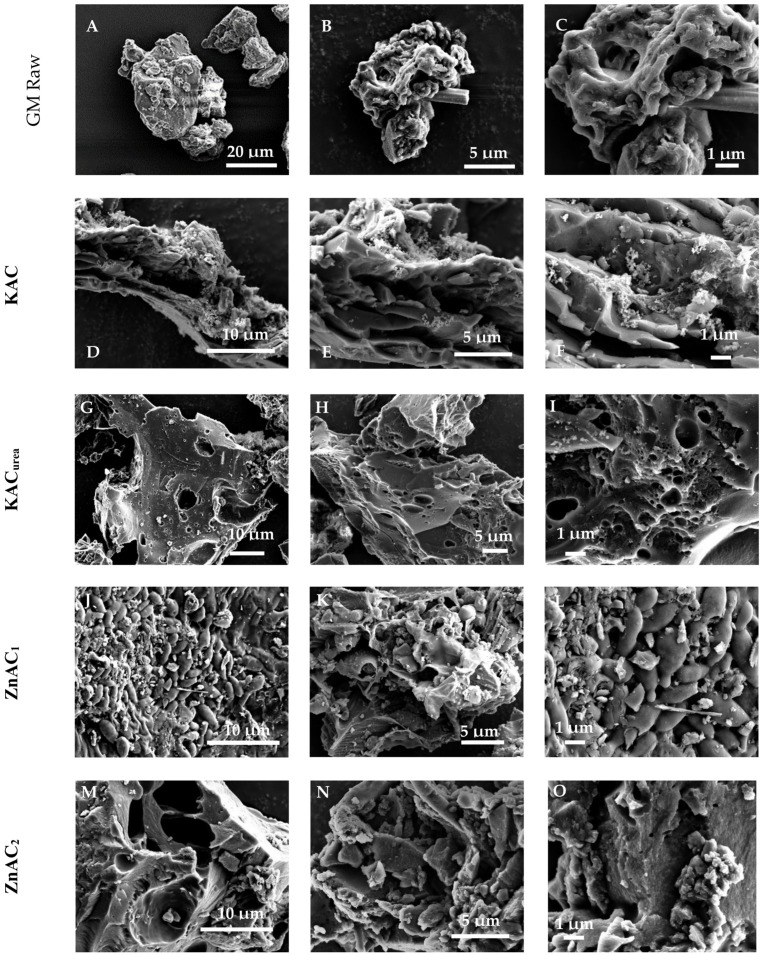
FESEM images of (**A**–**C**) GM precursor before any heat treatment, (**D**–**F**) KAC, (**G**–**I**) KAC_urea_, (**J**–**L**) ZnAC_1_ (**M**–**O**) ZnAC_2_.

**Figure 3 nanomaterials-12-01847-f003:**
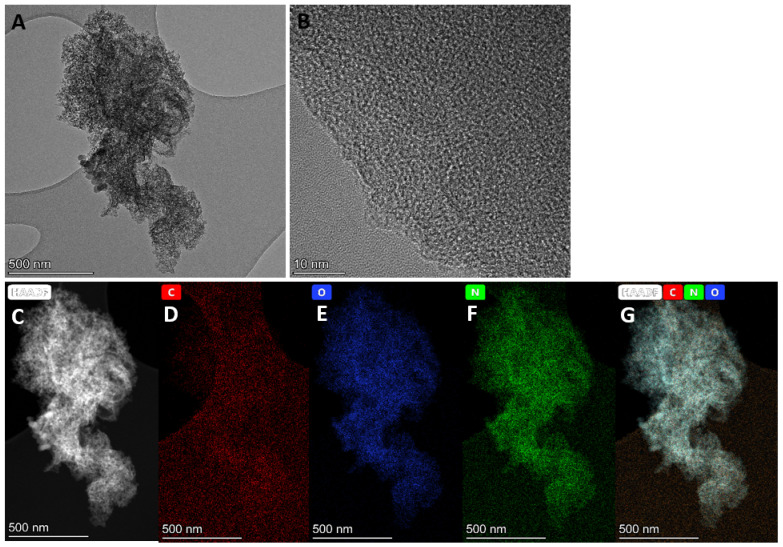
(**A**) Low-resolution and (**B**) high-resolution TEM images, (**C**) high-angle annular dark-field (HAADF) scanning transmission electron microscopy (STEM) image, and (**D**–**G**) energy-dispersive X-ray spectroscopy (EDS) elemental mapping of KAC_urea_.

**Figure 4 nanomaterials-12-01847-f004:**
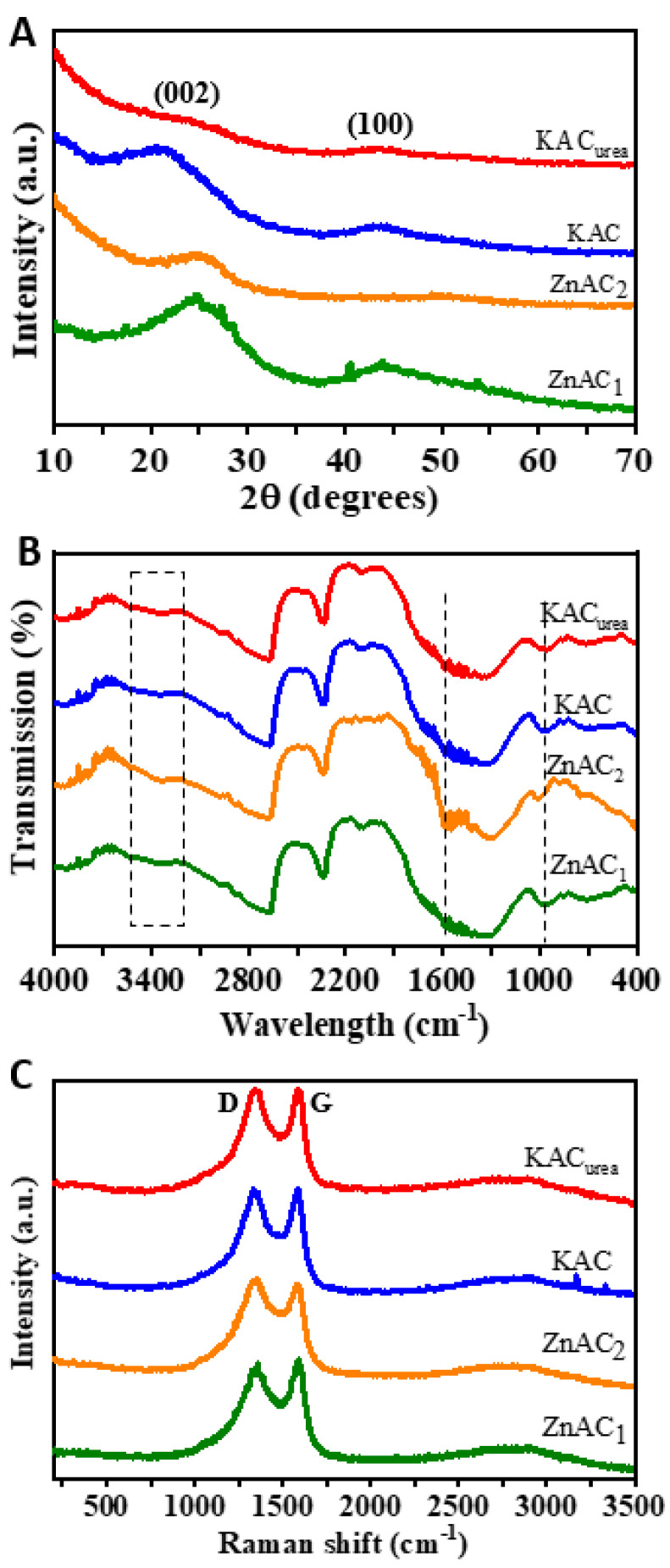
(**A**) XRD pattern, (**B**) FTIR spectra, and (**C**) Raman spectra of GM-derived AC, i.e., KAC_urea_, KAC, ZnAC_1_, and ZnAC_2_.

**Figure 5 nanomaterials-12-01847-f005:**
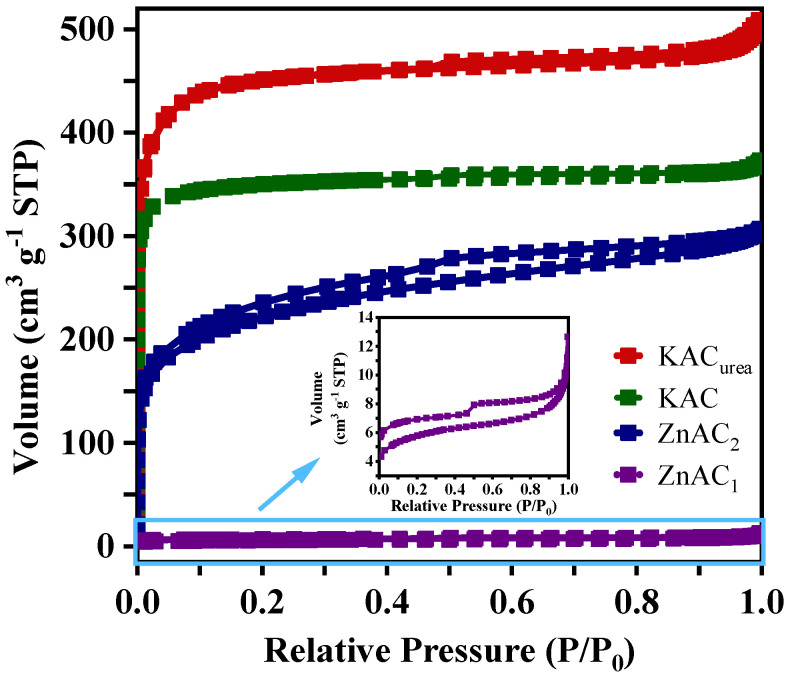
Comparison of N_2_ adsorption–desorption isotherms of KAC_urea_, KAC, ZnAC_1_, and ZnAC_2_. The inset shows the hysteresis loop for ZnAC_1_.

**Figure 6 nanomaterials-12-01847-f006:**
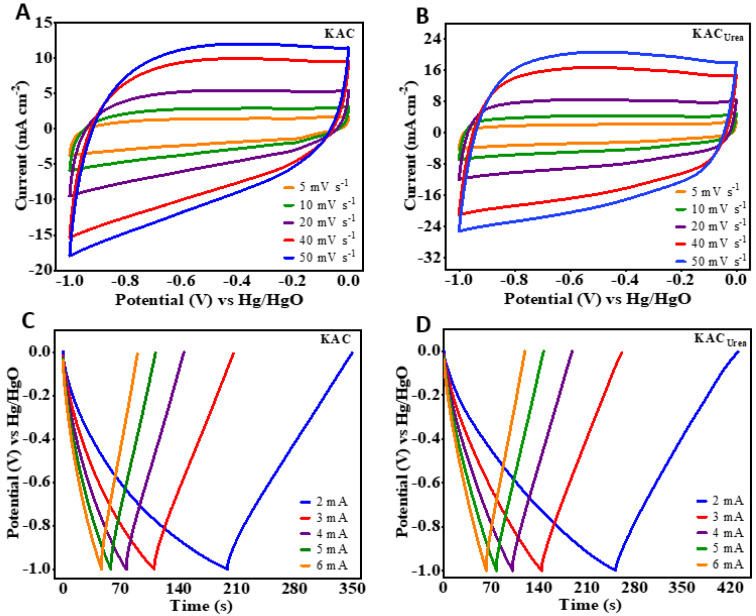
(**A**,**B**) Cyclic voltammetry (CV) and (**C**,**D**) galvanostatic charge–discharge (GCD) profiles of KAC and KAC_urea_.

**Figure 7 nanomaterials-12-01847-f007:**
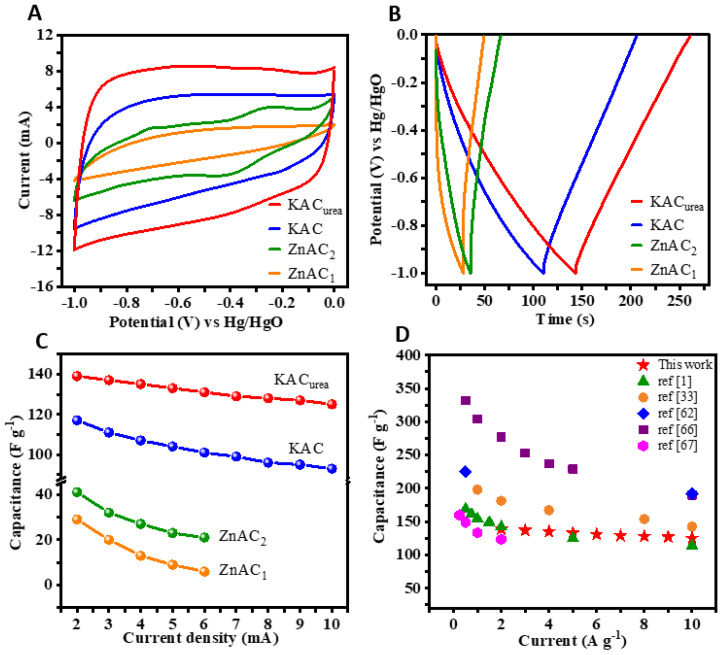
(**A**) CV at 20 mV s^−1^ and (**B**) GCD at 3 mA of GM-derived N-doped AC, tested in 2 M NaOH electrolyte. (**C**) Specific capacitances at varying applied currents of ZnAC_1_, ZnAC_2_, KAC, and KAC_urea_. (**D**) Capacitance comparison of the best-performing KAC_urea_ with the reported values in the literature [1,33,52,53,54].

**Figure 8 nanomaterials-12-01847-f008:**
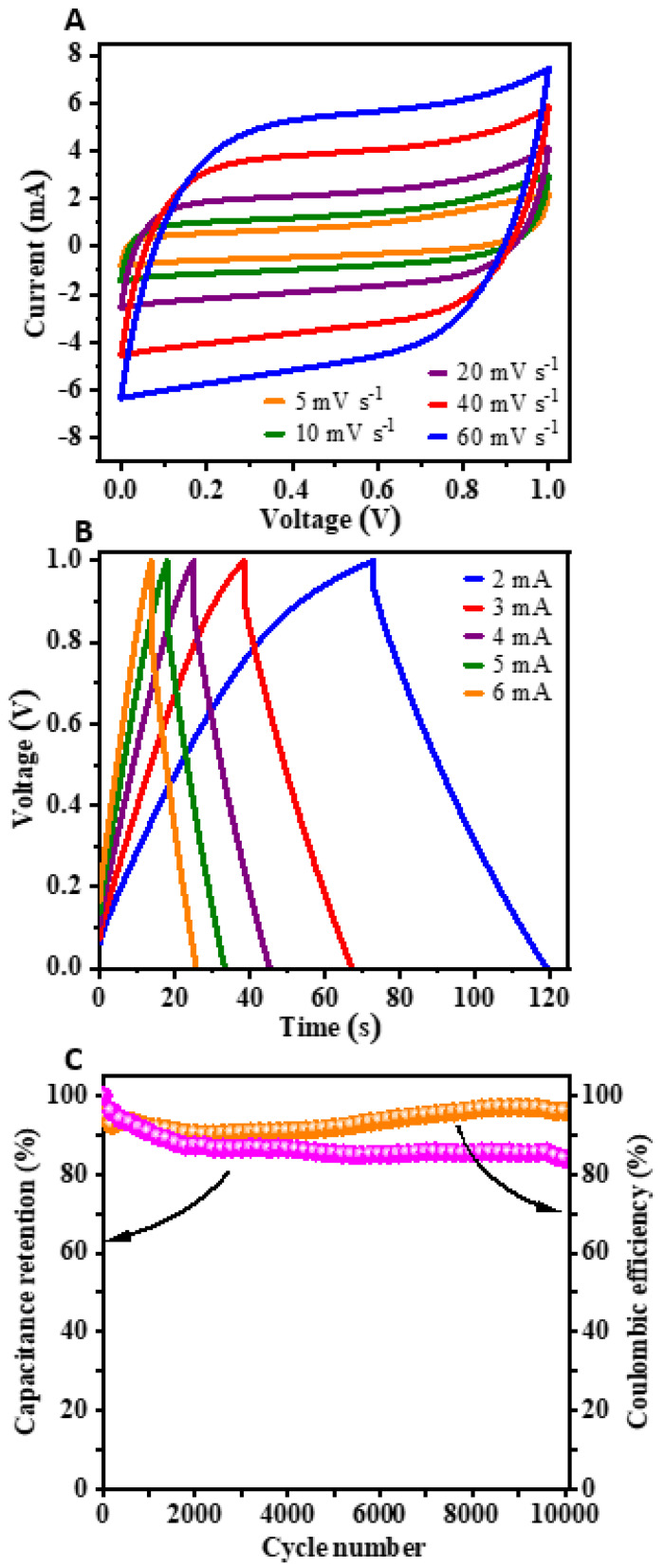
(**A**) CV at varying scan rates (5, 10, 20, 40, and 60 mV s^−1^) and (**B**) GCD profiles at varying current rates (2, 3, 4, 5, 6 mA) of the symmetric capacitor device (KAC_urea_/KAC_urea_) in an aqueous 2 M NaOH electrolyte with a 1 V window. (**C**) Capacitance retention and coulombic efficiency over 10,000 cycles.

**Figure 9 nanomaterials-12-01847-f009:**
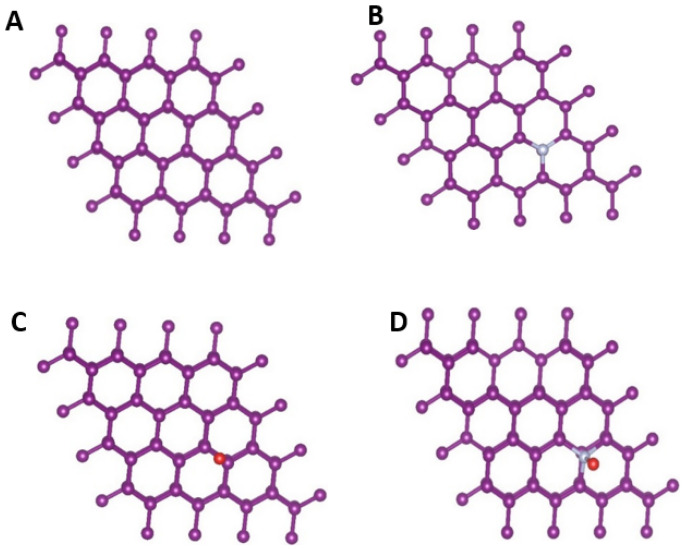
The optimized structures of graphite bilayers: (**A**) pristine graphite, (**B**) graphite substituted with a N atom on the surface, (**C**) pristine graphite with O adsorbate, and (**D**) graphite substituted with a N atom and O adsorbate. The figures were plotted using the VESTA software. The color code of the atoms is as follows—purple: C, white: N, red: O.

**Figure 10 nanomaterials-12-01847-f010:**
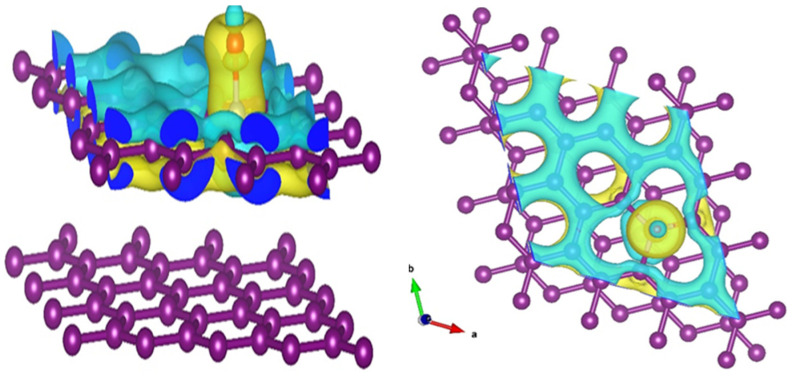
The charge density difference calculated for the N-doped graphitic surface with O adsorption. The pristine and N-doped graphite surfaces are also shown. The yellow and blue isosurfaces indicate charge gain and charge depletion, respectively. The isosurface level is 0.062368 e/Å3”. The color code of the atoms is as follows—purple: C, red: O, white: N.

**Figure 11 nanomaterials-12-01847-f011:**
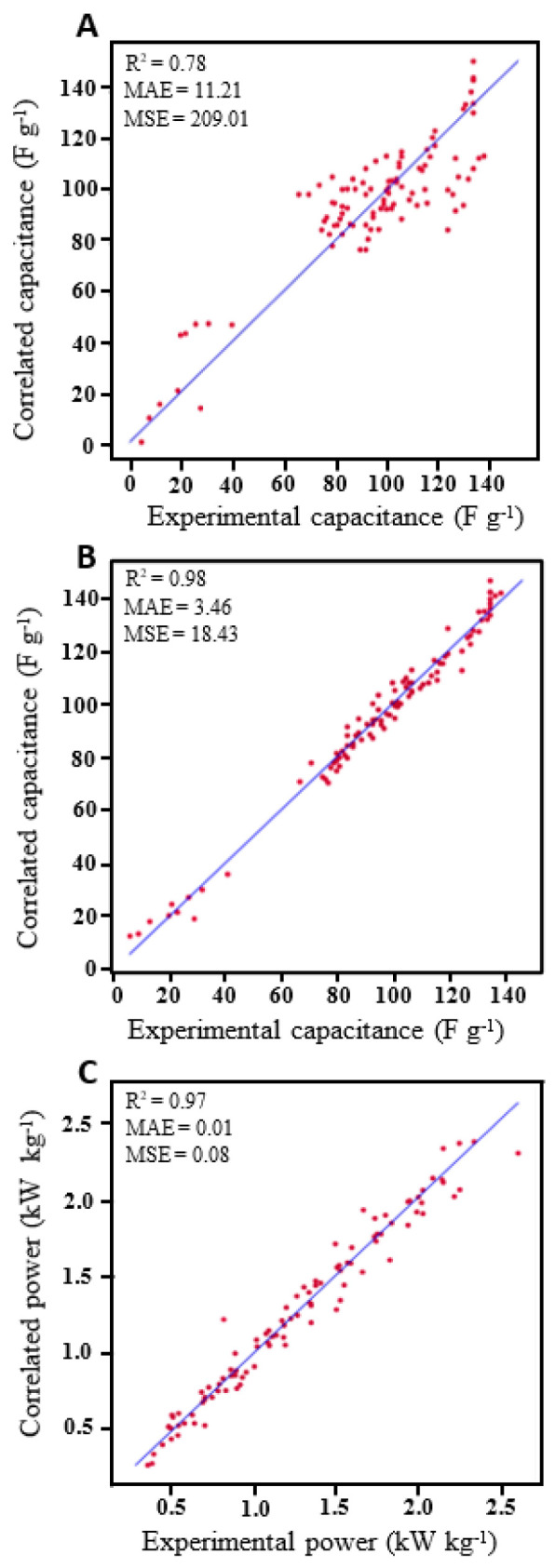
Comparison of correlation for the predicted specific capacitance (F g^−1^) and the actual experimental specific capacitance (F g^−1^) between the (**A**) MLP model with “6” inputs and (**B**) MLP model with more inputs; (**C**) correlated power (kW kg^−1^), and the actual power density (kW kg^−1^) of the samples.

**Table 1 nanomaterials-12-01847-t001:** Surface area and pore analysis parameters for the GM-derived AC materials.

Sample	BET Surface Area (m^2^ g^−1^)	t-Plot Micropore Area (m^2^ g^−1^)	Micropore Area (%)	Total Pore Volume (cm^3^ g^−1^)	t-Plot Micropore Volume (cm^3^ g^−1^)	Micropore Volume (%)
KAC_urea_	1356	1126	83.0	0.79	0.60	75.9
KAC	1128	998	88.5	0.58	0.53	91.4
ZnAC_1_	29	26	89.7	0.02	0.01	50.0
ZnAC_2_	711	275	38.7	0.46	0.15	32.6

**Table 2 nanomaterials-12-01847-t002:** Comparison of biomass-based carbon derived from various precursors, with their reported specific capacitances and other relevant parameters.

Ref.	Biomass Precursor	Activating Agent	Surface Area (m^2^ g^−1^)	Electrolyte	Current Density	Specific Capacitance (F g^−1^)
Wang et al. [1]	Apricot shell lignin	H_3_PO_4_	1474.82	6 M KOH	0.5 A g^−1^	169.05
Subramanian et al. [55]	Banana fibers	ZnCl_2_ KOH	1097 686	1 M Na_2_SO_4_	0.5 A g^−1^	74 66
Nabais et al. [56]	Coffee endocarp	CO_2_ KOH	709 361	1 M H_2_SO_4_	10 mA (~0.2 A g^−1^)	176 69
Gou et al. [57]	Wheat straw	KOH	772	6 M KOH	0.5 A g^−1^	226.2
Yan et al. [58]	Macadamia nutshell	KOH	2202	1 M Na_2_SO_4_	1 A g^−1^	155
Mondal et al. [49]	Shrimp shell	KOH	1271	6 M KOH	0.5 A g^−1^	239
Liu et al. [59]	Rice straw	KOH	1127	6 M KOH	1 A g^−1^	337
Huang et al. [52]	Wood sawdust	KOH	2294	6 M KOH	0.5 A g^−1^	225
Ramirez et al. [60]	Coffee husk	Steam K_2_CO_3_ KOH	1447 1156 2275	6 M KOH	0.5 A g^−1^	138 129 106
Xu et al. [61]	Rice straw	KHCO_3_	2786.5	6 M KOH	1 A g^−1^	317
Dubey et al. [62]	Human hair	KOH	1992.4	1 M H_2_SO_4_	1 A g^−1^	274.5
This work	GM	KOH	1356	2M NaOH	2 A g^−1^	139

**Table 3 nanomaterials-12-01847-t003:** Bond distances and Bader charges calculated for the pristine and N-doped graphite surfaces. Bader charges, calculated for the atoms nearest to the adsorbed O atom.

Configuration	Bond Length (Å)			Bader Charges		
	C-O	C-N	N-O	O	N	C
Pristine graphite + O	2.27			−0.35		
N-doped graphite		1.41			−2.66	0.98
						0.84
						0.84
N-doped graphite + O		1.49	1.40	−0.49	−1.21	0.69
						0.39
						0.39

## Data Availability

Not applicable.

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
