# Peer review of "Repurposing N-Doped Grape Marc for the Fabrication of Supercapacitors with Theoretical and Machine Learning Models"

_nanomaterials, 2022, doi:10.3390/nano12111847_

Round 1
Reviewer 1 Report
The manuscript "Repurposing N-doped Grape Marc for the fabrication of supercapacitors with theoretical and machine learning models' represents an interesting approach for actual sustainable research.
Nevertheless, some aspects have to be revised:
References are appropriately selected but each needs the associated DOI.
Some minor editing aspects (unexpected BOLD characters..., different size...downscripts...) have to be corrected
Author Response
Manuscript Number: nanomaterials-1732789
Responses to reviewer’s comments
Reviewer #1
Many thanks for taking the time to consider our manuscript and making possible these very constructive comments. Please find below a point-by-point response to the reviewer’s comments.
The manuscript “Repurposing N-doped Grape Marc for the fabrication of supercapacitors with theoretical and machine learning models” represents an interesting approach for actual sustainable research.
Nevertheless, some aspects have to be revised:
References are appropriately selected but each needs the associated DOI.
In the revised version of the manuscript, we have now included the DOIs for the references.
Some minor editing aspects (unexpected BOLD characters…, different sizes….down scripts…) have to be corrected.
Agreed. We thoroughly went through the manuscript and revised it. The changes are highlighted in the manuscript.

Reviewer 2 Report
Detailed work that fits well the scope of the journal. Good amount of data is presented, mainly materials science. DFT and ML are somewhat weak and need improvement.
- The machine learning models for verification purposes should be provided.
- The dataset for validation should be provided as a supporting information datafile.
- Root mean squared error (RMSE) should be reported instead of R2.
- For small datasets SVM, random forest and boosting algorithm are appropriate.
- The model is missing the repeatability measure, y-scrambling, cross-validation.
- Include hyperparameter tuning, including the optimization of the learning rate and algorithm, cost, and loss function, number of epoch etc.
- The application of machine learning in materials science is emerging, and previous diverse examples should be acknowledged.
- Greener preparation of AC electrodes are emerging and prior art should be mentioned (10.1021/acssuschemeng.6b02735; 10.1016/j.jclepro.2022.130922).
- The DFT description (both the method and the results) are too short and not placed into context.
Author Response
Manuscript Number: nanomaterials-1732789
Responses to reviewer’s comments
Reviewer #2
Many thanks for taking the time to consider our manuscript and making possible these very constructive comments. Please find below a point-by-point response to the reviewer’s comments.
- The machine learning models for verification purposes should be provided.
The ML model architecture has been provided in the earlier version itself. It is in the SI file.
“Section 2.5: Figure S1 presents the ML model architecture, which consists of three hidden layers with 40, 60, and 15 nodes.”
- The dataset for validation should be provided as a supporting information data file.
We agree. We have now provided the dataset in the excel sheets.
- Root mean squared error (RMSE) should be reported instead of R2.
Sure but in our case to evaluate how well the chosen variables of a model can explain the variation in the predictor variables in percentage terms, we have employed R2. We have reported a strong linear relationship between the two variables and giving the highest R2 of 0.97. Therefore, we have followed and reported coefficient of determination (R2) values in this work.
RMSE may be a misleading indicator of average error in our studies with limited sample numbers. Hence, mean squared error (MAE) has been reported in this work as a better metric.
Nevertheless, the values are quoted in the manuscript. The following lines are included in the manuscript.
“In addition, the RMSE value has decreased from 14.45 to 4.29 indicating the importance of the additional parameters in modeling the system. The improved model has been used to predict the power density as in Figure 10C, which gave 0.97 for R2 and a value of 0.28 for RMSE.”
- For small datasets SVM, random forest and boosting algorithm are appropriate.
We agree. However, this is only a preliminary work carried out for the ML modelling. We will take your suggestion for our future studies.
It should be also noted that from the output of the Pearson correlation test it can be noticed that many non linear relationships exist between the input and output parameters. In such a situation, the best way to model a system is to use a non-linear predication model such as MLP.
- The model is missing the repeatability measure, y-scrambling, cross-validation.
Please refer the SI section for the details provided on the cross validation, etc. As per the best practice used in the ML field, a 5 fold cross validation scheme was used. The 5 - fold cross-validation splits the data into 5 equal-sized blocks and one block for testing and the other blocks for training for the model. The results presented in this section consider the average of those 5 - fold cross-validation.
- Include hyperparameter tuning, including the optimization of the learning rate and algorithm, cost and loss function, number of epoch etc.
Please refer the SI section for the details provided. The key inputs employed in the MLP model are given in Table S1.
"The relu’ and ‘linear’ activation functions are used for the model, and the optimizer used is ‘adam’. The MLP was trained for a maximum iteration of 2500 for this model. Out of the experimental data, 80% was used for training, and 20% was used to test the MLP model"
- The application of machine learning in materials science is emerging, and previous diverse examples should be acknowledged.
Sure, we agree. The following text has been included in the revised version.
“Machine learning prediction has also been widely adopted in the field of material science 63, 64 . Depending on its end applications, ML has been classified into material property prediction, the discovery of novel materials, and various other purposes like battery management, etc.. When constructing the ML models for material property predictions, the researchers have extended their insights on various aspects, such as predicting the electronic properties of inorganic crystals65, geometric features of gas storage and separation using metal-organic frameworks66, transport properties in granular materials67, and the effect of the pore size on capacitance68. Besides, the ML models are also used in efficient optimization by designing their parameters69, applying them to DFT problems 70, monitoring the state of the battery during operation71, etc. However, surface morphology has not been considered in previous ML studies. The particle morphologies can be quite diverse based on biomass precursor or synthesis conditions that affect the energy storage performance.72-74 “
(63) Liu, Y; Zhao, T.; Ju, W.; Shi, S. Materials discovery and design using machine learning, J. Materiomics. 2017, 3, 159 -177. https://doi.org/10.1016/j.jmat.2017.08.002.
(64) Schmidt, J.; Marques, M.R.G.; Botti, S.; Marques, M.A.L. Recent advances and applications of machine learning in solid-state materials science, Npj Comput. Mater. 2019, 5, 83. https://doi.org/10.1038/s41524-019-0221-0.
(65) Isayev, O.; Oses, C.; Toher, C.; Gossett, E.; Curtarolo, S.; Tropsha, A. Universal fragment descriptors for predicting properties of inorganic crystals, Nat. Commun. 2017, 8, 1–12. https://doi.org/10.1038/ncomms15679.
(66) Fernandez, M.; Woo, T.K.; Wilmer, C.E.; Snurr, R.Q. Large-scale quantitative structure-property relationship (QSPR) analysis of methane storage in metal-organic frameworks, J. Phys. Chem. C. 2013, 117, 7681–7689. https://doi.org/10.1021/jp4006422.
(67) Van Der Linden, J.H. Narsilio, G.A. Tordesillas, A. Machine learning framework for analysis of transport through complex networks in porous, granular media: A focus on permeability, Phys. Rev. E. 2016, 94, 1–16. https://doi.org/10.1103/PhysRevE.94.022904.
(68) Fang, Y.; Zhang, Q.; Cui, L. Recent progress of mesoporous materials for high performance supercapacitors, Microporous Mesoporous Mater. 2021, 314, 110870. https://doi.org/10.1016/j.micromeso.2020.110870
(69) Diao, Y.; Yan, L.; Gao, K. A strategy assisted machine learning to process multi-objective optimization for improving mechanical properties of carbon steels, J. Mater. Sci. Technol. 2022, 109, 86–93. https://doi.org/10.1016/j.jmst.2021.09.004.
(70) Snyder, J.C.; Rupp, M.; Hansen, K.; Müller, K.R.; Burke, K. Finding density functionals with machine learning, Phys. Rev. Lett. 2012, 108, 1–5. https://doi.org/10.1103/PhysRevLett.108.253002.
(71) Shi, S.; Gao, J.; Liu, Y.; Zhao, Y.; Wu, Q.; Ju, W.; Ouyang, C.; Xiao, R. Multi-scale computation methods: Their applications in lithium-ion battery research and development, Chinese Phys. B. 2015, 25, 018212. https://doi.org/10.1088/1674-1056/25/1/018212 .
(72) Chong, S.; Lee, S.; Kim, B.; Kim, J. Applications of Machine Learning in Metal-Organic Frameworks. Coord. Chem. Rev. 2020, 423, 213487. https://doi.org/10.1016/j.ccr.2020.213487
(73) Ge, M.; Su, F.; Zhao, Z.; Su, D. Deep Learning Analysis on Microscopic Imaging in Materials Science. Mater. Today Nano 2020, 11. https://doi.org/10.1016/j.mtnano.2020.100087
(74) Krishnamurthy, D.; Weiland, H.; Barati Farimani, A.; Antono, E.; Green, J.; Viswanathan, V. Machine Learning Based Approaches to Accelerate Energy Materials Discovery and Optimization. ACS Energy Lett. 2019, 4 (1), 187–191. https://doi.org/10.1021/acsenergylett.8b02278
- Greener preparation of AC electrodes are emerging and prior art should be mentioned (10.1021/acssuschemeng.6b02735, 10.1016/j.jclepro.2022.130922)
Sure, we agree. The following text has been included in the revised version.
“On the other hand, environmentally friendly biomass precursors with green approaches to synthesizing porous carbon materials are also seen to be emerging, which can provide pragmatic improvements in the sustainable processing of energy-storing materials. Goldfarb et al. [9] proposed an integrated process for extracting biofuel from the pyrolysis of pistachio nutshell and impregnating it with KOH activation processes to produce activated carbon. This integrated process increased the biofuel yield up to 25%, and the AC was used for electrochemical capacitor studies. McNair et al. [18] reported green binders (γ-valerolactone) and solvents (cellulose acetate, carboxymethyl cellulose) as alternatives to replacing the conventional solvents (N-methyl-2-pyrrolidone; NMP) and binders (fluorinated such as PVDF) for membrane capacitive deionization electrodes with enhanced capacitive performance. This work aimed to reduce the environmental impacts and chemical consumption used in electrode processing. All these studies conclude sustainable biomass electrodes could be biodegradable.”
J.L. Goldfarb, G. Dou, M. Salari, M.W. Grinstaff, Biomass-Based Fuels and Activated Carbon Electrode Materials: An Integrated Approach to Green Energy Systems, ACS Sustain. Chem. Eng. 5 (2017) 3046–3054. https://doi.org/10.1021/acssuschemeng.6b02735.
McNair, G. Szekely, R.A.W. Dryfe, Sustainable processing of electrodes for membrane capacitive deionization (MCDI), J. Clean. Prod. 342 (2022) 130922. https://doi.org/10.1016/j.jclepro.2022.130922.
- The DFT description (both the method and the results) is too short and not placed into context.
We agree. In the revised version of the manuscript, the following section is added with two additional figures to support the content.
“The electronic structure analysis is carried out to understand the hybridization of orbitals at the surface. The density of states (DOS) of the pristine, as well as the N-doped graphite surface, is presented in Fig. S6. In the presence of N atom, the total DOS at the Fermi level increases indicating that the chances of hybridization with an adsorbate increase.
We have calculated the difference in charge density caused by the adsorption of O at the N-doped graphitic surface. Figure 10 shows the charge density difference plot. The yellow and blue isosurfaces indicate charge gain and depletion, respectively. It can be seen that charge gain at the site of N doping is larger, while the doped O atom also gained charge partially. The C atoms on the surface, which are directly bonded with the N atom lose charge and a charge reconstruction occurs at the surface, which helps for the enhanced adsorption of the O atom. The charge density difference analysis and calculated Bader charges agree with the shortened bond distances in presence of N doping as presented in Table 3.
End of comments

Round 2
Reviewer 2 Report
Most of the comments were addressed.